# The PROMISES study: a mixed methods approach to explore the acceptability of salivary progesterone testing for preterm birth risk among pregnant women and trained frontline healthcare workers in rural India

Danielle Ashworth [1], Pankhuri Sharma,[2] Sergio A Silverio [1], Simi Khan,[2] Nishtha Kathuria,[2] Priyanka Garg,[2] Mohan Ghule,[2] V B Shivkumar,[3] Atul Tayade,[4] Sunil Mehra,[5] Poonam V Shivkumar,[6] Rachel M Tribe,[1] The wider PROMISES Study Team

DA and PS are joint first authors.

For numbered affiliations see end of article.

**Correspondence to**
Dr Rachel M Tribe;
rachel.tribe@kcl.ac.uk

## ABSTRACT

**Introduction** India has an overall neonatal mortality rate of 28/1000 live births, with higher rates in rural India. Approximately 3.5 million pregnancies in India are affected by preterm birth (PTB) annually and contribute to approximately a quarter of PTBs globally. Embedded within the PROMISES study (which aims to validate a low-cost salivary progesterone test for early detection of PTB risk), we present a mixed methods explanatory sequential feasibility substudy of the salivary progesterone test.

**Methods** A pretraining and post-training questionnaire to assess Accredited Social Health Activists (ASHAs) (n=201) knowledge and experience of PTB and salivary progesterone sampling was analysed using the McNemar test. Descriptive statistics for a cross-sectional survey of pregnant women (n=400) are presented in which the acceptability of this test for pregnant women is assessed. Structured interviews were undertaken with ASHAs (n=10) and pregnant women (n=9), and were analysed using thematic framework analysis to explore the barriers and facilitators influencing the use of this test in rural India.

**Results** Before training, ASHAs' knowledge of PTB (including risk factors, causes, postnatal support and testing) was very limited. After the training programme, there was a significant improvement in the ASHAs' knowledge of PTB. All 400 women reported the salivary test was acceptable with the majority finding it easy but not quick or better than drawing blood. For the qualitative aspects of the study, analysis of interview data with ASHAs and women, our thematic framework comprised of three main areas: implementation of intervention; networks of influence and access to healthcare. Qualitative data were stratified and presented as barriers and facilitators.

**Conclusion** This study suggests support for ongoing investigations validating PTB testing using salivary progesterone in rural settings.

### Strengths and limitations of this study

► This is the first study to present a mixed methods approach to understanding the acceptability of salivary progesterone testing for risk of preterm birth in rural India.

► This study presents data supporting the introduction of a previously unused point of care test which can easily and quickly be used outside of formal healthcare settings and delivered by community healthcare workers.

► This study presents and integrates mixed methods data from both women and community healthcare workers in a rigorous and methodical way.

► This study was conducted in a low-resource setting and demonstrates how a simple intervention can provide the possibility to improve the prenatal care of women and their babies.

## INTRODUCTION

Globally, India's contribution to neonatal mortality is highest in the world with an estimate of 0.75 million[1] neonatal deaths in 2013. India's neonatal mortality rate is 28/1000 live births, and in rural India, the situation is even graver with 31 neonatal deaths per 1000 live births.[2] Preterm birth (PTB) has been identified as one of the significant causes of neonatal deaths both in the world[3] and in India.[1] Approximately 3.5 million pregnancies in India are affected by PTB annually,[4–6] many infants die and those surviving often live with disability. Globally, PTBs are the highest in India (23.4% of all PTBs).[1 7] This burden is further affected by the low awareness level and utilisation of health services, an issue that

is exacerbated in rural India. Knowledge of, and around, PTB is limited even among community level health workers. This lack of knowledge hinders identification of mothers at risk of PTB and can delay access to healthcare services to manage such pregnancies. Considering the high burden of PTB and lack of knowledge thereof, an innovative study with an aim to validate a low-cost salivary progesterone test (PROMISES) is being conducted for early detection of risk of PTB among pregnant women in two rural districts of India.[8]

Even today, many myths and misconceptions prevail about pregnancy and childbirth in rural areas, mainly due to inadequate formal education, poor accessibility to healthcare services and lack of trained frontline health workers. Moreover, the major role of decision-making about healthcare of expectant mothers is mainly done by their immediate family members. The recent national survey in India indicates that only 11.4% of women in rural India alone makes decision about their healthcare (National Family Health Survey 4, India, 2015-16; NFHS-4). These contextual factors play a major role in implementing effective innovative interventions for improving maternal and neonatal health.

In rural India, government-instituted community healthcare workers, Accredited Social Health Activist (ASHAs), play a significant role in linking pregnant women to maternal and child healthcare services.[9] These frontline health workers work in their residential communities/villages as health activists, educators and providers of basic essential services. ASHAs provide several support services to pregnant women[10] and invest considerable effort in identifying and building trust with expectant mothers and their family members. A multitude of evidence[11 12] suggests that these links with frontline health workers have improved maternal and neonatal health interventions. For ASHA workers, the required minimum formal education[13] is only up to class eight; hence, it is important to provide adequate knowledge and regular trainings to ASHAs on topics of maternal and neonatal health in order to improve practice and sustainability of innovative interventions in these fields.

Hence, as part of the PROMISES study[8] two of the aims were: (a) to assess the knowledge and educate the frontline health workers, ASHAs, on PTB and salivary progesterone sampling through a training programme and (b) to consider the feasibility and acceptability to women and health workers, of using salivary progesterone in the rural settings for its further scale-up as a 'point of care test'. More specifically, the research objectives were to determine:

► If training frontline health workers improved their knowledge on PTB and salivary progesterone sampling.
► Whether salivary progesterone PTB tests were acceptable to frontline health workers and pregnant women.
► The range of facilitating factors and barriers influencing the use of salivary progesterone PTB tests in Indian rural settings.

## METHODS
### Study design and setting

A mixed methods study was undertaken to explore knowledge of PTB in frontline healthcare workers, and the acceptability and feasibility of salivary progesterone testing through a survey and structured interviews, respectively. This work formed part of a prospective study evaluating the feasibility and accuracy of saliva progesterone test to predict PTB, entitled—'Low-cost salivary progesterone testing for detecting the risk of preterm births in rural community settings of India, The PROMISES study'.[8] Participating healthcare workers were government-instituted community healthcare workers, ASHAs, whose primary role is health education and promotion of good health practice and health service accessibility and utilisation. ASHAs were instrumental in recruitment of pregnant women for the PROMISES study, and thus, capacity building through training was essential for effective implementation. Furthermore, the study explored the acceptability of salivary progesterone test among pregnant women eligible for PROMISES.

We conducted a pre–post training assessment of ASHAs knowledge of PTB; structured interviews with a sample of pregnant women and ASHAs discussing acceptability and feasibility of salivary progesterone testing for PTB; and a cross-sectional survey of pregnant women recruited to PROMISES which included questions on acceptability of salivary progesterone testing.

This study was carried out in the state of Madhya Pradesh, India, within two districts Panna and Satna, selected because of their high-fertility and neonatal mortality rates.[14 15] A 1-day training programme was conducted for ASHAs at eight primary healthcare centres (PHCs). Overall four 1-day training programmes were conducted, three at primary health centres and one at a district field office. Training included information about the maternal and child health services within the health systems, outreach services, high-risk pregnancies, newborn care, PTBs and general diagnostic services in pregnancy. They were then introduced to the PROMISES study and salivary progesterone as a screening tool for PTB. All interviews with ASHAs and pregnant women were conducted in primary health centres.

### Recruitment and data collection

All the participants provided their written informed consent. Both ASHAs and women were identified through their involvement in and eligibility for the main PROMISES study; details of inclusion/exclusion criteria can be found in the study protocol.[8]

### Pre–post training questionnaire: ASHAs

A questionnaire of 25 items was designed to collect information on ASHAs expertise, employment, work experience and general knowledge of PTB and neonatal care. The questionnaire was administered to 201 ASHAs recruited in the PROMISES study before and after

**Table 1** Geographical and demographical characteristics of the ASHAs undertaking the preterm birth training programme

| Variable (n=201) | N | % |
|---|---|---|
| District | | |
| Panna | 112 | 55.7 |
| Satna | 89 | 44.3 |
| Years of education | | |
| ≤5 years | 7 | 3.5 |
| 6–9 years | 68 | 33.8 |
| 10–14 years | 110 | 54.7 |
| ≥15 years | 16 | 8 |
| Mean (SD) | 10.15 (2.32) | |
| Years of experience as an ASHA | | |
| ≤4 years | 37 | 18.4 |
| 5–8 years | 70 | 34.8 |
| 9–12 years | 94 | 46.8 |
| Mean (SD) | 7.50 (3.11) | |

ASHAs, Accredited Social Health Activists.

training within both districts (n=112 from Panna, n=89 from Satna).

## Cross-sectional questionnaire: women

Survey of the first 400 pregnant women participating in PROMISES and providing a saliva sample exploring women's experiences of salivary progesterone testing and its acceptability by the women (n=184 from Panna, n=216 from Satna).

## Interviews: pregnant women and ASHAs

Through opportunity sampling, ASHAs (n=10, n=5 from Panna, n=5 from Satna) and pregnant women (n=9, n=5 from Panna, n=4 from Satna; one participant from the Satna region withdrew) were interviewed by two multilingual interviewers (authors PS, SK) who were part of the research team. Interviews were face to face, structured and informational using a topic guide developed using

the research aims, the experiences of the research team and information received from the field team. Interviews explored the health services provided to pregnant women, the role of ASHAs in caring for pregnant women and managing PTB, and women's thoughts on PTB and their experiences of salivary progesterone testing including feasibility and any challenges of this test within the rural setting. Participants were offered travel compensation and refreshments as reimbursement for their time. Interviews were audio-recorded (with the participants' consent), transcribed verbatim and translated into English by PS and SK.

## Patient and public involvement

We did not directly include patient and public involvement (PPI), but the study rationale was presented and discussed with our UK National Institute for Health Research-recognised PTB PPI group, and the acceptability of the research was informed by participant feedback from a similar UK-based study. Due to the participatory nature of this research, this study provides important PPI from women and community healthcare workers in rural India who are rarely consulted about research.

## Data analysis

Quantitative data were uploaded and managed within the online secure database, MedSciNet. Descriptive analysis of questionnaire data was conducted in IBM SPSS V.24 with results expressed as counts and percentages. Knowledge before and after training was compared by the McNemar test and a value of 0.05 considered to indicate statistical difference.

Preliminary analysis of the 19 interviews was undertaken by a member of the main PROMISES team (PS) with further detailed analysis by two other study team members (SAS, DA). Codes and generated themes were cross-checked with members of the wider PROMISES team. A thematic framework analysis was employed[16] focusing on the perceptions of the intervention, the influencing factors which affected women's engagement with the service and the function and implementation of the intervention. The framework was devised and agreed

**Table 2** ASHA's knowledge of preterm birth (PTB) before and after training programme

| Question (n=201) | Before training | | After training | | P value |
|---|---|---|---|---|---|
| | N | % | N | % | |
| PTB identified as a cause of newborn death | 97 | 48.3 | 196 | 97.5 | 0.000 |
| Had heard of PTB | 151 | 75.1 | 193 | 96.0 | 0.000 |
| Correctly defined PTB by gestation (<37 weeks) | 49 | 24.4 (32.5*) | 78 | 38.8 (40.2*) | 0.253 |
| Correct knowledge of potential risk factors for PTB | 10 | 5.0 (6.6*) | 84 | 41.8 (43.3*) | 0.000 |
| Had any awareness of a test for PTB | 0 | 0.0 | 122 | 60.7 (62.9) | . |
| Correctly identified additional postnatal support needed for PTB | 31 | 15.4 (21.8*) | 135 | 67.2 (69.6*) | 0.000 |

*Proportion with missing values excluded in the denominator.
ASHAs, Accredited Social Health Activists.

(SAS, DA) using the overarching research questions, the interview schedules and notes made when familiarising with the interview transcripts.

In-line with a multidisciplinary approach to thematic framework analysis,[17] interviews were first read in their entirety by three analysts to achieve familiarity with the interview content. One of the analysts had undertaken some of the interviews (PS), while the other two had not been involved in the data collection (DA, SAS). Interviews were then uploaded into NVivo for data management and analysis. Coding was focused using the framework to guide analysis. Theme names within the framework were adapted and refined as the analytical processes went on. Analysis was completed in a way which met both public and global health needs while being sensitive to the socioeconomic and cultural environment in which these participants resided.[18–20] Theme saturation was reached after analysis of approximately 60% of transcripts had been analysed,[21] which although achieved with relatively few transcripts, is unsurprising given the structured nature of the interviews.[22] For each of the main themes, data were charted and matrices used to analyse patterns, sorting by participant status (ie, ASHAs or women) and divided into barriers and facilitators. The most eloquent example quotations have been selected to represent the themes and analysis.

## RESULTS

Two hundred and one ASHAs completed the training programme. Table 1 shows the geographical and demographical characteristics of ASHAs undertaking the training programme, of the 201 ASHAs, 56% were from the Panna district and 44% from Satna district. Although there was a range in experience levels of the ASHAs, on average they had a high level of education and advanced experience.

### Pre–post training programme questionnaire: ASHAs

Before the 1-day training programme, approximately three quarters of the ASHAs had heard of PTB; however, their knowledge of the risk factors, causes and postnatal support required for PTB was very limited (table 2). After the training programme, there was a significant improvement in the ASHAs' knowledge of PTB compared with before training, with a 21% increase in the proportion of ASHAs that had heard of PTB, 49% increase in knowledge that PTB can cause newborn death, 52% increase in additional postnatal support required after PTB, and 37% increase in correct knowledge of PTB risk factors. Before the training programme, none of the 201 ASHAs were aware of potential tests for PTB, this is compared with 61% after training.

Over half of the ASHAs reported they had experience of caring for women with PTB (table 3); however, their experience of providing women with information on PTB showed that over a quarter of the women are not interested in the information provided and over 90% of those

**Table 3** ASHA's experiences of caring for women with preterm birth (PTB)

| Question (n=201) | Before training | |
|---|---|---|
| | N | % |
| Had dealt with PTB to date | 114 | 56.7 (75.5)* |
| What are your experiences (barriers) in providing information to women for preterm birth? | | |
| Women are not interested | 47 | 23.4 (31.8)* |
| Family members are not interested | 135 | 67.2 (91.2)* |
| Other | 8 | 4.0 (5.4)* |

*Proportion with missing values excluded in the denominator.
ASHAs, Accredited Social Health Activists.

who answered this question said family members were not interested in the information.

### Cross-sectional questionnaire: pregnant women

Table 4 shows the geographical and demographical characteristics of the first 400 women to provide a saliva sample within the PROMISES cohort. The average age was 24 years with 8 years of education.

Women recruited to the PROMISES prospective cohort completed sections of a PROMISES questionnaire which captured information on women's sociodemographic and lifestyle characteristics, reproductive history and saliva sampling. Within this questionnaire, women were asked three questions about the acceptability of the salivary test. These three questions have been extracted for the first 400 participants, of the 2000 cohort (table 5). Despite all 400 women reporting the salivary test was acceptable and 82% finding the test easy, 84% reported the test was not better than drawing blood and 98% did not think the test was quick. What women disliked about the test was not clear as only three women reported they disliked associated mouth dryness, gagging or any embarrassment.

### Interviews: pregnant women and ASHAs

Following analysis of interview data with ASHAs and women, our thematic framework comprised of three main areas: implementation of intervention; networks of influence and access to healthcare (tables 6 and 7). Contextual data were also captured as were details of local or traditional practices of, and attitudes towards, medicine, PTB and pregnancy ailments. This was not used for analysis but instead for information to set the scene in which the data were collected.

The first area, implementation of intervention, covered information about women's perceived acceptability (degree of tolerance) of the intervention, as well as the feasibility and usefulness. The second area, networks of influence, was made up of themes of local knowledge, family and healthcare professional influences, and wider support networks. These forms of knowledge were noted as being transferred from the influencers and networks to women. Thirdly, access to healthcare comprised distance

**Table 4** Geographical and demographical characteristics of first 400 women to have a saliva sample recruited to promises cohort

| Variable/question (n=400) | N | % |
|---|---|---|
| District | | |
| Panna | 184 | 46 |
| Satna | 216 | 54 |
| Maternal age (years) | | |
| 18–20 | 62 | 15.5 |
| 21–25 | 251 | 62.8 |
| 26–30 | 79 | 19.8 |
| >30 | 8 | 2 |
| Mean (SD) | 23.6 (3.1) | |
| Years of education | | |
| ≤4 | 47 | 11.8 |
| 5–8 | 191 | 47.8 |
| 9–12 | 131 | 32.8 |
| >12 | 31 | 7.8 |
| Mean (SD) | 7.8 (3.8) | |
| Monthly income* (rupees) | | |
| Less than 2500 | 261 | 62.3 |
| 2501–5000 | 116 | 29 |
| 5001–7500 | 12 | 3 |
| 7501–10 000 | 7 | 1.8 |
| 10 001 and above | 4 | 1 |
| Have you ever smoked or used tobacco in any form? | | |
| Yes and still use | 17 | 4.3 |
| Yes, gave up before pregnancy | 7 | 1.8 |
| No | 376 | 94 |
| Do you consume any drink containing alcohol? | | |
| Yes | 1 | 0.3 |
| No | 399 | 99.8 |

*As of 2020, 100 Indian rupees = US$1.41/£1.08 sterling.

**Table 5** Acceptability of salivary progesterone test to the first 400 pregnant women recruited to the promises cohort

| Question (n=400) | N | % |
|---|---|---|
| Did the participant find the salivary test acceptable? | | |
| Yes | 400 | 100 |
| No | 0 | 0 |
| No opinion | 0 | 0 |
| Did the participant feel the salivary test was: better than drawing blood? | | |
| Yes | 65 | 16.3 |
| No | 335 | 83.8 |
| Easy to give? | | |
| Yes | 325 | 81.3 |
| No | 75 | 18.8 |
| Quick? | | |
| Yes | 9 | 2.3 |
| No | 391 | 97.8 |
| Convenient help to know the risks involved? | | |
| Yes | 0 | 0 |
| No | 400 | 100 |
| None of the above | | |
| Yes | 11 | 2.8 |
| No | 389 | 97.3 |
| Other | | |
| Yes | 0 | 0 |
| No | 400 | 100 |
| What did the participant dislike about the salivary test: mouth dryness? | | |
| Yes | 7 | 1.8 |
| No | 393 | 98.3 |
| Time taken? | | |
| Yes | 1 | 0.3 |
| No | 399 | 99.8 |
| Gag reflex? | | |
| Yes | 2 | 0.5 |
| No | 398 | 99.5 |
| Embarrassment? | | |
| Yes | 0 | 0 |
| No | 400 | 100 |
| None of the above? | | |
| Yes | 391 | 97.8 |
| No | 9 | 2.3 |
| Other | | |
| Yes | 0 | 0 |
| No | 400 | 100 |

and cost of travel, transport issues as well as geographical factors.

## Implementation of intervention

Implementation of intervention included two main subthemes: women's perceived acceptability (degree of tolerance) towards the potential intervention and the usefulness and feasibility. The intervention was perceived as acceptable by both ASHAs and pregnant women with 50% of ASHAs and 78% of women stating facilitating factors. This was compared with approximately 10% of both ASHAs and women who raised issues which suggested the intervention was not acceptable. No ASHAs or women raised issues in relation to the usefulness and feasibility of the intervention; however, 20% of ASHAs and 44% of women highlighted factors which could be codified as useful and feasible.

## Networks of influence

Networks of influence was the largest theme and contained four main subthemes: healthcare professional influences, family influences, wider support networks and local knowledge. Healthcare professional influence was

**Table 6** Thematic framework analysis of interviews with pregnant women

|  |  | Implementation of intervention | Networks of influence | Access to healthcare |
|---|---|---|---|---|
| Women | Facilitators | Woman 4 (Panna)**:** "I had no problem when I went for sonography. When I gave my sample, I asked outreach worker that why she needs my sample than she informed that they would examine it to determine the health of my child."<br><br>Woman 8 (Satna): "I assume that women would agree to cooperate if they are told that it is for their child well-being. We, who have undergone this process can also convey and then I don't think women would hesitate."<br><br>Woman 9 (Panna): "My saliva sample was collected at my house and the process was convenient to me." | Woman 4 (Panna): "Outreach worker approached me along with ASHA and took me to district hospital. My husband stays in Panna and he comes to district hospital whenever I go for check-up."<br><br>Woman 5 (Satna): "Usually it is either of my mother-in-law and sister-in-law accompanies me to sub-centre… I had no problem in travelling… I got to know about preterm birth from outreach worker, she told me that baby who is born preterm is not healthy. It would be good to know more about pre-term birth."<br><br>Woman 6 (Satna): "I will take care of my nutrition, as I do not want my child to born undernourished. Preterm birth can lead to under-nourishment of child. He remains weak. I have seen one example, there was a child who was born preterm and when he was growing, his legs were paralysed." | Woman 1 (Panna): "On my third visit, my sonography was conducted. As such, I did not face any difficulty… we commuted by bus."<br><br>Woman 8 (Satna): "My husband accompanies me to the health centres, so I face no problem in travelling. Public transport is also safe." |
| | Barrier | Woman 7 (Satna): "Not all pregnant women would agree for providing their saliva, as many do not even go for sonography" | Woman 7 (Satna): "They want to go [for sonography] but their families do not allow them. They do not have faith on government hospitals and say that they will go to the private hospitals for check-up." | Woman 3 (Panna): "Travelling is problem during the rainy season as roads are not very good."<br><br>Woman 6 (Satna): "I went for sonography twice, first I went to government hospital which is 20–25 km away and I waited for long time and doctor left for the day. Again, when I went to District hospital my sonography was not conducted. Then my in-laws suggested me to go to private hospital. Moreover, government hospitals reports are not reliable. So, I went to private hospital for my sonography."<br><br>Woman 9 (Panna): "I faced no difficulties in saliva collection, but the ultrasound was a time taking procedure where I had to travel to a distance of more than 60 km and it took the whole day." |

ASHA, Accredited Social Health Activist.

**Table 7** Thematic framework analysis of interviews with ASHAs

| | | Implementation of intervention | Networks of influence | Access to healthcare |
|---|---|---|---|---|
| ASHA | Facilitators | ASHA 3 (Panna): "Women are also very easily motivated for collecting saliva and for going for ultrasound." ASHA 8 (Satna): "Women are convinced to give saliva samples more as it is non-invasive and is for their benefit." | ASHA 5 (Panna): "I accompany the project's outreach workers during saliva sample collection in order to motivate the pregnant women. Sometimes I accompany the pregnant women for ultrasound also as their families' requests to." ASHA 10 (Satna): "Sometimes I also [accompany] pregnant women for ultrasounds as their family members are more comfortable in sending their females with us rather than the outreach workers." | ASHA 4 (Panna): "The women prefer going with me for ultrasound testing and are also able to commute by the auto rikshaws." |
| | Barrier | ASHA 2 (Panna): "Since most women here are uneducated with high parity, so it is very difficult to make them understand. It is mostly their families who stop. They say what has to happen will happen, they already have 3–4 kids before as well." | ASHA 1 (Panna): "It was easy to convince the pregnant women and family for ultrasound scan except in few cases where they were daily wagers and refused to go for ultrasound scan." ASHA 9 (Satna): "Pregnant women (are) willing to go for ultrasounds but sometimes family members refuse for this as they feel that it is not required." | ASHA 1 (Panna): "The pregnant women deny [travel] for getting their ultrasound done as they are daily wagers." ASHA 2 (Panna): "Those women who are uneducated and poor they do not go for sonography. Even if tell them to go they would make some excuse. In government hospital if they go, most of the times it is closed and then these women do not go again......... One woman went for sonography, her sonography was not done that day, and now she refuses to go again." ASHA 5 (Panna): "Travelling is the major problem in going to primary healthcare centre for regular antenatal clinic. Many women also refuse due to lack of conveyance. Thus, many times I bring women on my own" |

ASHAs, Accredited Social Health Activists.

mainly perceived as a transfer of knowledge from ASHAs and other healthcare professionals to pregnant women and was seen to be facilitatory in nature by women (78%) but not necessarily by ASHAs (20%). Despite this, only 10% of ASHAs and 22% of women suggested healthcare-related knowledge transfer barriers. The next subtheme was the family influence. Within this subtheme, 10% of ASHAs and approximately 30% of women suggested these to be facilitatory with 50% of ASHAs and 22% of women suggesting family influence acting as a barrier to their care. The third subtheme was wider support networks which was never seen as a facilitator. Analysis showed this subtheme was only raised by women (approximately 30%) as a barrier influencing their care. A final subtheme of local knowledge was seen by only ASHAs to pose some

barriers to pregnant women accessing healthcare. In this respect, 30% of ASHAs confirmed these barriers, whereas 20% of ASHAs and 22% of women considered local knowledge as a facilitator instead.

### Access to healthcare
Access to healthcare was made up of three subthemes: distance and cost of travel; mode of transport and geographical factors.

Clearly distance was only ever seen as a barrier to accessing healthcare, with 20% of ASHAs and approximately 30% of women raising it as an issue. Furthermore, transport was seen as a barrier to accessing healthcare by 30% of ASHAs and 11% of women. Similar numbers (10% of ASHAs and 22% of women) found transport which

could meet their needs (though this may be a product of the study whereby ASHAs were offering transport as part of the study involvement). Finally, geographical factors were only occasionally mentioned, these were exclusively raised as barriers by women (22%).

## DISCUSSION

This study presents an analysis of the feasibility and acceptability of using salivary-based tests for predicting PTB and the training of community health workers to facilitate the sampling. It has successfully demonstrated support for the research questions posed. A major strength of this study was the mixed methods approach implemented, whereby we were able to triangulate quantitative and qualitative data from both women and ASHAs. Mixed methods data triangulation can be undertaken in a number of different ways, and this study employed a parallel data analysis plan.[23] Parallel data analysis is where collection and analysis of both quantitative and qualitative data sets are carried out separately, and findings are not compared until the interpretation stage.[24 25]

In terms of healthcare worker training, it was clear that although ASHAs understood the concept of PTB (57% had experience of caring for women with PTB), their baseline knowledge of the condition was limited despite over 80% of ASHAs reporting 5 or more years of work experience and the standard government-issued training which is provided to all ASHAs.[26] ASHAs understood PTB was a cause of neonatal death but did not have clarity over the gestation cut-off as a definition for PTB. Results of the questionnaire showed a lack of understanding by ASHAs of antenatal risk factors, antenatal tests and postnatal support for PTB and preterm babies. This was further evidenced in both interview data collected from ASHAs. This is unsurprising as although community health workers have become a more integral part of the health workforce in low-income and middle-income countries (LMICs),[27] basic health literacy remains low.[28] In spite of this, previous literature has suggested that although health literacy may be universally poor, the work of community health workers is vitally important for hard-to-reach groups or those with high levels of social complexity.[29–31]

The 1-day training sessions enabled ASHAs to have a greater understanding of PTB. As training evaluation was conducted directly after the training session, this only provides evidence of short-term recall. Follow-up of retention of information learnt and ability to apply knowledge would have been useful, though fell outside of the remit of this study. However, ASHAs were involved in the PROMISES study for approximately 24 months with continued contact with study field staff and pregnant women. As ASHAs played a key role in explaining the PROMISES study to women during the consent process, it is likely that knowledge was retained and that involvement in the study had a longer lasting impact on their knowledge of PTB as has been found in similar studies with community

health workers who are engaged in research on pregnancy and/or women's health.[32 33]

A key goal of the study was to determine, in advance of potentially validating a saliva progesterone as a predictor of PTB for women in rural India, the acceptability of the test within this setting. Saliva sampling, while not as invasive a blood sampling, has been disliked by women in the UK for various reasons due to embarrassment and dryness of mouth.[34] Reassuringly, both quantitative and qualitative data from women and ASHAs in the PROMISES study show the saliva progesterone testing to be acceptable. This is useful information, not only for our proposed progesterone test, but also for any other future point-of-care biomarker test.

Furthermore, although we did not formally measure health literacy in women participating in the PROMISES study, previous studies have shown that women's involvement in health-related research can promote better maternal health literacy,[35 36] which may be a more holistic, if not anticipated, benefit of this study.

### Strengths and limitations

A major strength of this study is the fact we present a mixed methods approach to understanding the acceptability of a novel intervention into a hard-to-reach (rural; socio-economically deprived) population in an LMIC using community health workers. Mixed qualitative and quantitative methodological approaches have been hailed as providing the most viable way to identify and answer clinically relevant research questions.[37] Furthermore, the study presents data supporting the introduction of a previously unused point-of-care test which can easily and quickly be used outside of formal healthcare settings, making this an important advance in prenatal care for rural populations in LMICs who may not have access to transport to services or to the healthcare facilities themselves (due to financial constraints, poor weather or infrastructure, or family influence). However, like all feasibility studies, we have identified certain limitations. Most notably, the qualitative data collected were assessed[38] as being of poor quality due to the high level of shadow data (ie, participants talking about other people's experiences). We compensate for this by using a rigorous thematic framework methodological approach which enabled us to interrogate the data and distil it into coherent themes, while providing a percentage cover of all the topics raised in our framework for both women and ASHAs. Additionally, the opportunity sample could be seen as a methodological weakness. While the use of the qualitative data has helped maximise our understanding of the situation in rural India and helped to explain the quantitative findings, more could be done to enhance the generalisability of similar research in the future.[39] For example, collecting relevant demographical information on each pregnancy (eg, parity, multiple pregnancy, maternal age) may enable future qualitative analyses to stratify by these variables which may give more in depth insight. On consideration, these limitations

are outweighed by the strengths of a study designed in a setting low in resources to demonstrate how simple interventions provide the possibility to improve the prenatal care of women and their babies.

## Conclusion and future work

In summary, these data provide support for our ongoing study of the feasibility of using salivary progesterone testing for prediction of PTB in rural settings. In addition, it highlights the potential usefulness of saliva for additional biomarker developments. The study further provides good evidence for this type of test being useful to shift cultural narratives to allow women more agency during pregnancy; however, there is still a need for women to have better access to routine antenatal care even if these simple interventions can be carried out outside of formal healthcare settings. Although this study provides evidence for sample collection in homes and then be returned to central facilities for testing in addition to successful mechanisms for reporting results, ideally this test should be undertaken in conjunction with routine antenatal care at healthcare facilities and so women should be afforded the time and opportunity to safely access these services.

**Author affiliations**
[1]Department of Women & Children's Health, King's College London, London, UK
[2]Research and Innovation Unit, Mamta Health Institute for Mother and Child, New Delhi, Delhi, India
[3]Department of Pathology, Mahatma Gandhi Institute of Medical Sciences, Sevagram, Maharashtra, India
[4]Department of Radio-Diagnosis, Mahatma Gandhi Institute of Medical Sciences, Sevagram, Maharashtra, India
[5]Young People and Sexual and Reproductive Health and Rights Unit, Mamta Health Institute for Mother and Child, New Delhi, Delhi, India
[6]Department of Obstetrics and Gynecology, Mahatma Gandhi Institute of Medical Sciences, Sevagram, Maharashtra, India

**Acknowledgements** We would like to thank all the women who took part in this study, the ASHAs and the clinical staff who performed the ultrasound scans.

**Collaborators** The wider PROMISES Study Team includes: Paul T Seed (Department of Women & Children's Health, King's College London); Ritu Dargan (Obstetrics and Gynecology, Mamta Health Institute for Mother and Child) and Archana Sarkar, Lalita Sengupta, Vikram Singh (Research and Innovation Unit, Mamta Health Institute for Mother and Child).

**Contributors** Conceptualisation: (RMT, PVS, PS, SK, MG, SM and the wider PROMISES Study Team); methodology: (SAS, DA, PS, SK, MG and RMT); software: (SAS and DA); validation: (RMT and PS); formal analysis: (DA, SAS, PS); investigation: (DA, SAS); resources: (RMT, PVS, PS, SK, NK, PG, MG, VBS, AT and SM); data curation: (PS, DA, RMT and SAS); writing: original draft: (DA, PS, SAS and RMT); writing: review and editing: (NK, PVS, PS, SK, NK, PG, MG, VBS, AT, SM, SAS, DA and RMT); visualisation: (SAS); supervision: (RMT, PVS, SM and MG); project administration: (RMT, PVS, PS, SK, NK, PG, MG, VBS, AT and SM); funding acquisition: (PVS, RMT, VBS, SM and the wider PROMISES Study Team).

**Funding** This study was conducted as a part of project titled 'Low-cost salivary progesterone testing for detecting the risk of preterm births in rural community settings of India'. Funding for this project was provided by the Grand Challenge India, 2015—All Children Thriving—India partners (Biotechnology Industry Research Assistance Council; Department of Biotechnology, India and the Bill and Melinda Gates Foundation) for its overall implementation.

**Competing interests** SAS (King's College London) is supported by the National Institute for Health Research Applied Research Collaboration South London (NIHR ARC South London) at King's College Hospital NHS Foundation Trust. The views expressed are those of the authors and not necessarily those of the NIHR or the Department of Health and Social Care.

**Patient consent for publication** Not required.

**Ethics approval** Ethical approval was obtained from both Mamta Health Institute for Mother and Child (MAMTAHIMC) Ethical Review Board (MERB/Dec-2016/002) and the Institutional Review Board of Mahatma Gandhi Institute of Medical Science (MGIMS/IEC/OBGY/289/2016) prior to any recruitment. Written informed consent was obtained from pregnant women and ASHAs.

**Provenance and peer review** Not commissioned; externally peer reviewed.

**Data availability statement** Data are not available to be shared.

**ORCID iDs**
Danielle Ashworth http://orcid.org/0000-0002-3977-4260
Sergio A Silverio http://orcid.org/0000-0001-7177-3471

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
