## [Reviewer comments · BMJ Open]

ARTICLE DETAILS

TITLE (PROVISIONAL)	The PROMISES Study: A mixed methods approach to explore the acceptability of salivary progesterone testing for pre-term birth risk amongst pregnant women and trained frontline healthcare workers in rural India.
AUTHORS	Ashworth, Danielle; Sharma, Pankhuri; Silverio, Sergio; Khan, Simi; Kathuria, Nishtha; Garg, Priyanka; Ghule, Mohan; Shivkumar, V; Tayade, Atul; Mehra, Sunil; Shivkumar, Poonam; Tribe, Rachel M.

VERSION 1 – REVIEW

REVIEWER	Della Vedova, A. M. University of Brescia, Italy
REVIEW RETURNED	17-Jun-2020

GENERAL COMMENTS	Thank you very much for the opportunity of revising the manuscript entitled: "The PROMISES Study: Impact of training, research involvement, and salivary progesterone testing on women and frontline health workers' understanding of preterm birth." I found the methodological structure of the article excellent. I greatly appreciated the article that used a mixed methods approach to understanding the acceptability of a novel intervention, contextually to the evaluation of the feasibility and the usefulness of using salivary progesterone testing for prediction of preterm birth. The research also demonstrates the effectiveness of training health workers for prevention in perinatal health and collects the real experiences of women and health workers towards the intervention. This is a valuable way of verifying how the recipients of an intervention are involved and perceive it. It is a research intervention that used in an elegant and clear way a complex methodology and, with its results, will help reduce the preterm birth rate in the most at-risk populations and improve the prenatal care of women and their babies. I recommend the article for publication. I have found some small aspects that can be made clearer: - Abstract: "The deaths associated with preterm birth in India contribute a quarter of the global preterm related deaths". At first reading, it seems to me not very clear that India contributes to a quarter of the cases worldwide. Pag. 8 line 10-11: "Maternity care in rural India is heavily influenced by a number of a number of social, familial, and lifestyle factors." Perhaps it would be better to check the wording of this sentence.
--

REVIEWER	Dr Bola Grace University College London
REVIEW RETURNED	09-Oct-2020

GENERAL COMMENTS	Comments to the authors I appreciate the opportunity to review your manuscript. Your paper aimed to assess frontline health workers' knowledge of preterm birth and salivary progesterone sampling; understand efficacy of the training provided; to explore women's attitudes towards the use of a new saliva based point of care diagnostic test. This is part of a larger study which was conducted to validate salivary progesterone as biomarker for an alternative rapid diagnostic test for preterm birth in India. I commend you for focussing on women in a country where the incidence of PB is high, for speaking directly to frontline healthcare workers and the sample representative of the women for which the product is designed, and for using mixed methods. Your paper is well written and succinct – thank you. I have some concerns on with your methods section which I hope you can clarify. I enjoyed reading your discussion and confusions. Please see below for my detailed comments. Introduction:  • 8: Minor: any data more recent than 2013? • Rephrase lines 20 – you don't mean pregnancies die, and 21 - Globally PTB deaths are the highest in India? Methods I struggled with this section as I feel some important details are missing.  • How did you recruit? Did you have a recruitment plan? • Which platform did you use? • How did you screen participants? • What were your inclusion/exclusion criteria? • What was your dropout rate? • Did you incentivise? If so, did you consider the implications on your study findings? • Was there a topic guide? How was it developed? • How many interviewers? Did they use the same topic guide? • Did you translate or were the interviews conducted in English? What are the implication (e.g. data integrity)?  • Line7: Did you use a validated questionnaire? Results  • Well written. • Can you add brief descriptors/ criteria to your quotes? Discussion  • Good justification of study limitation. • Any other pointers in terms of generalisability of your study? Thanks again for the opportunity and all best wishes with your manuscript!
--

VERSION 1 – AUTHOR RESPONSE

Reviewer(s)' Comments to Author:

Reviewer: 1

Abstract

1. "The deaths associated with preterm birth in India contribute a quarter of the global preterm related deaths". At first reading, it seems to me not very clear that India contributes to a quarter of the cases worldwide.

This sentence has been clarified/edited.

2. Pag. 8 line 10-11: "Maternity care in rural India is heavily influenced by a number of a number of social, familial, and lifestyle factors." Perhaps it would be better to check the wording of this sentence.

The Key Messages sections in which this sentence was situated has been removed from manuscript as per the editor's comments.

Reviewer: 2

Introduction:

1. 8: Minor: any data more recent than 2013?

Additional, more recent, references have been added to the Introduction.

2. Rephrase lines 20 – you don't mean pregnancies die, and 21 - Globally PTB deaths are the highest in India?

This sentence has been clarified/edited.

Methods

3. How did you recruit? Did you have a recruitment plan?

4. Which platform did you use?

5. How did you screen participants?

6. What were your inclusion/exclusion criteria?

7. What was your dropout rate?

8. Did you incentivise? If so, did you consider the implications on your study findings?

9. Was there a topic guide? How was it developed?

10. How many interviewers? Did they use the same topic guide?

11. Did you translate or were the interviews conducted in English? What are the implication (e.g. data integrity)?

12. Line7: Did you use a validated questionnaire?

Comment 4 - Quantitative data was recorded on paper and entered onto the main study's secure database (medscinet). Qualitative data was audio-recorded (with the participants' consent), transcribed verbatim and translated into English and analysis conducted in NVivo – see 'Data analysis' and 'Recruitment & data collection' sections of the manuscript.

Comments 3 & 5-11 - Additional details of recruitment plan, eligibility (screening, inclusion/exclusion), dropout rates, incentivisation, topic guide, number of interviewers and translation have been added to the 'Recruitment & Data collection' section of the Methods to address these comments.

Comment 12 - The questionnaire was developed for the purposes of evaluating a training programme for the PROMISES study and thus was not validated.

Results

4 Can you add brief descriptors/ criteria to your quotes?

Additional details of the location of the pregnant women and ASHA's have been added to Table 6 and 7.

Discussion

5 Good justification of study limitation.

Thank you.

6 Any other pointers in terms of generalisability of your study?

An additional explanation of generalisability has been added to the 'Strengths and Limitations' section of the Discussion.

VERSION 2 – REVIEW

REVIEWER	Bola Grace University College London
REVIEW RETURNED	30-Nov-2020
GENERAL COMMENTS	Thank you for your effort. I am happy with your corrections. Best wishes with your manuscript